# The Voice of Teachers about School and the Teaching Profession in Guinea-Bissau

**Filipe Dias** * and **Maria José Rodrigues**

CIEB—Investigation Center for Basic Education, Polytechnic Institute of Bragança, 5300-253 Bragança, Portugal; mrodrigues@ipb.pt

\* Correspondence: filjmdias@hotmail.com

**Abstract:** Teachers' view of the school is determinant for their action and condition of their activity and their professional development. This study focuses on the voice of teachers at community management schools in Guinea-Bissau in order to know the vision they have of the school and the teaching profession. For data collection, a personal and professional characterization questionnaire and interviews with thirty elementary school teachers were used. The results show that teachers value continuing education as a tool that allows them to perform the profession better. They identified some constraints related to the difficulty in the Portuguese language, the lack of didactic materials and teachers, isolation, and lack of community participation. Teachers refer to the school as a place of transmission of knowledge and values, highlighting the school as the key to development. According to teachers, the need for more infrastructure and materials as well as the greater involvement and commitment of the educational community are the main factors that must be improved. To conclude, we highlight the role that teacher training plays in the quality of their work and the efforts necessary for schools to respond to the needs of society.

**Keywords:** school; community schools; Guinea-Bissau; teachers

## 1. Introduction

Currently, in school as well as in society, the acquisition of skills and abilities that allow an integral and healthy development of individuals is valued so that they feel good about themselves and their environment. There are several studies that have been carried out on the school and on the role of its actors for the development of a quality education complying with the guidelines of any country regarding its educational system. Overall, schools have the responsibility to take a more active role in promoting social justice and in training individuals capable of dealing with and responding to the needs of the ever-changing society. In this sense, Freire [1] highlighted that the school should be a space for the formation of autonomous subjects capable of thinking critically and acting in a transformative way in society, considering that the school should be a space of liberation in which students are encouraged to think critically and question the established norms and values. Thus, the school organization is a social unit subject to a process of historical construction, thus presenting itself as loaded with meanings.

In Guinea-Bissau, the educational system comprises, sequentially, pre-school, basic, secondary, technical-vocational, and higher education [2]. This study focuses on teachers of basic education, which, according to the referenced document, is universal and mandatory. However, access to basic education is still limited for many children, especially in rural areas [3]. According to the World Bank [4], one of the main barriers to access to basic education is the lack of adequate schools and infrastructures. Many schools are small and cannot afford the number of children who attend them. In addition, most schools have few pedagogical resources and do not have books, teaching materials, or adequate sanitary facilities, which hinders the teaching and learning process [5].

Another challenge of the education system in Guinea-Bissau is the lack of qualified teachers [6]. Many teachers are poorly paid and do not have access to initial or continuous pedagogical training, leading to a disinvestment in their professional development. This leads to a lack of motivation and commitment, which undermines the quality of education [7]. The country's political and economic instability also affects basic education [8]. Frequent changes of government and political tensions disrupt the education system and affect the motivation of students and teachers [7].

In the same line of thought, Gomes [9] stated that basic education in the country still has many limitations, such as the lack of adequate infrastructure, the lack of trained human resources, and the lack of effective public policies focused on education. The author emphasized the need for the greater participation of civil society and the state in promoting significant changes in the country's education system.

In rural communities, these problems take on a greater dimension in addition to the difficulties of communication with the central educational authorities, which makes it difficult to obtain support and resources to improve teaching conditions; all of this characterizes a very weak educational system [10]. According to UNICEF [10], in 2016, only about 47% of school-aged children in rural areas were enrolled in the first cycle of basic education compared to 87% in urban areas. In addition, primary school completion rates in rural areas were only 43% compared to 70% in urban areas.

Based on these premises, this study aims to know the vision that teachers of basic education have about school and the teaching profession. This research intends to elucidate the importance that teachers attribute to their profession, the constraints inherent to their work, the functions that the teachers attribute to the school, the importance of the training of teachers for their work, and, finally, how the school should be in the opinion of the teachers. According to Mainardes et al. [11], teachers' views of school can be influenced by several factors, such as working conditions, salary, professional recognition, and continuing education, among others. These aspects can directly influence the motivation and quality of the teaching and learning process. In addition, it is important to highlight that teachers' views of the school can also be influenced by educational policies and dominant discourses in society.

The methodology used for data collection is presented below. Next, we present the analysis and discussion of the results. In the end, some conclusions and considerations are elaborated that include the limitations of this study and recommendations for future study.

## 2. The Research Methodology

Regarding the nature of the research, it is a case study inserted in a qualitative approach of a descriptive and interpretative character [12] because it is intended as an approach that seeks to understand and interpret the meaning attributed to certain realities. According to Denzin and Lincoln [13], this approach aims to explore the experiences of the participants in the investigation of their own cultural and social context, with the purpose of understanding the meaning that people attribute to these experiences. Creswell [14] added that the qualitative approach allows researchers to obtain detailed information about participants' experiences and perspectives to understand the context and meaning of those experiences. The case study format was chosen because it aims to explore a system that is limited in time and depth through the collection of data from multiple sources of information rich in context [14] and because it is the most appropriate research strategy to know the "how" and "why" of a current event over which the researcher has little or no control [15].

In this study, we listened to 30 teachers of the 1st cycle of basic education who work in 17 rural community management schools in the Contuboel sector, located in the region of Bafatá, in the east of Guinea-Bissau. This group of teachers was chosen because it is a group of teachers that is part of a network of community management schools supported by an international NGO. At the same time, these teachers have biweekly meetings for training and planning classes, which facilitated the formation of the focus groups for the interviews. The fact that the group of teachers belong to a network of schools and meet

every second week facilitated access to teachers and data collection. These schools are definitive construction schools with two classrooms and basic equipment. In addition, each school has didactic materials distributed by the NGO that supports them. We tried to understand the meaning these teachers attribute to their experiences and how these experiences are shaped by the social and cultural contexts in which they occur.

For data collection, we used a questionnaire to characterize the participants and the semi-structured interviews conducted in two focus group to hear the perspective of teachers about the role of the school and their profession. According to Morgan [16], a focus group is a guided discussion that involves a group of people with similar experiences and perspectives on a particular topic or problem. The objective of the focus group is to obtain in-depth information about the participants' perceptions of the topic of interest, allowing the researcher to better understand the beliefs, attitudes, and behaviors related to that topic. According to Silva et al. [17], the focus group allows an interactive group configuration in which participants can talk openly with other members of the group. The interaction between the group participants can lead to discussions and reflections that would not be obtained through individual interviews, which is why it was the technique chosen in this case. However, we must consider that the conclusions obtained from a focus group are specific to the group and cannot be generalized to the population as a whole.

The data obtained in the focus group were transcribed and later subjected to content analysis, which allowed the identification and categorization of thematic patterns in the transcribed text [18]. Content analysis allowed us to understand the complexity of the data at a deeper level [18]. Content analysis involved a subjective process, which is subject to the interpretation of researchers, as Bruner argued [19].

## 3. Results and Discussion

### 3.1. Characterization of Teachers (Gender, Age, Academic Training, Length of Service, Means of Transport, and Distance to School)

Of the 30 teachers participating in the study, 83.3% are male. Regarding age, 50% of teachers are between 31 and 45 years old. The distribution of ages can be seen in Figure 1 in more detail.

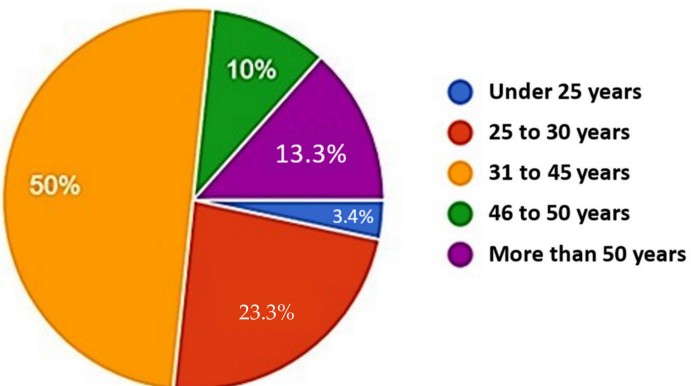

**Figure 1.** Distribution of teachers' ages.

Regarding the academic training of the teachers, it is very variable: one of them completed the 6th year of basic school, while the others have a bachelor's degree. None of the professors has a master's degree or higher. Figure 2 shows the diversity of the academic qualifications of the teachers interviewed. It should be noted that only 30% of the teachers interviewed have initial pedagogical training, but none of them acquired this training before starting the teaching functions.

Regarding the length of service, the responses are distributed over the intervals shown in Figure 3. Most teachers (53%) have less than 10 years of experience.

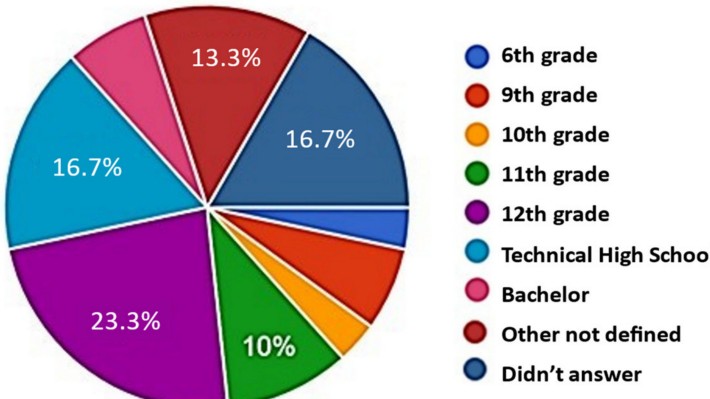

**Figure 2.** Academic training of teachers.

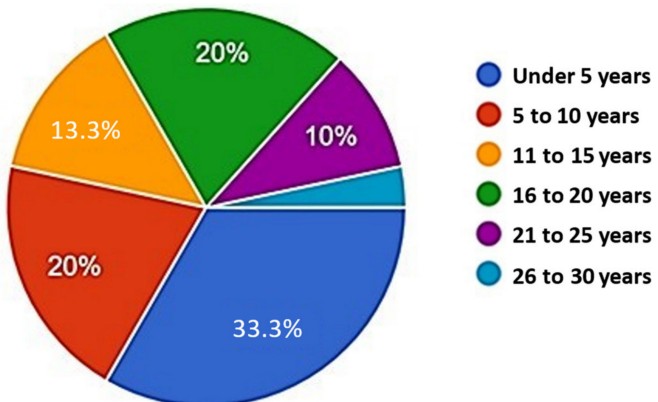

**Figure 3.** Teachers' length of service.

As for the employment relationship with the state, 83% of the teachers have some link with the state, while 17% are community teachers. About 60% of teachers have management functions in the school, such as that of principals or sub-principals. All teachers attend or have attended continuing teacher-education courses taught by various entities, as shown in Figure 4. The international NGO AIDA (*Ayuda, Intercambio, y Desarrollo*) (82%) and FEC (*Fundação Fé e Cooperação*) (14%) jointly occupy the first place for entities that provided continuous training to the teachers who participated in this study.

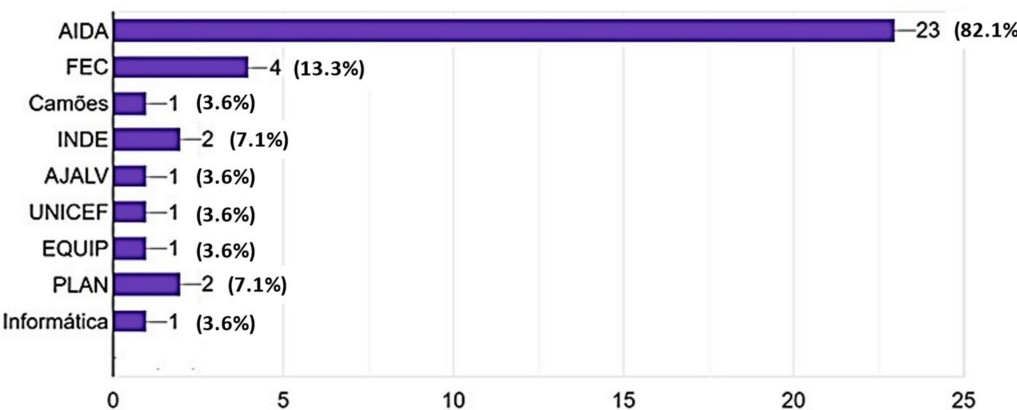

**Figure 4.** Teacher-training entities.

Regarding the distance that the teachers have to travel from their residence to the school where they work, it was verified that most of teachers (63%) live in the community where they work, while the others live more than 5 km from the school and travel by bicycle and motorcycle (Figure 5).

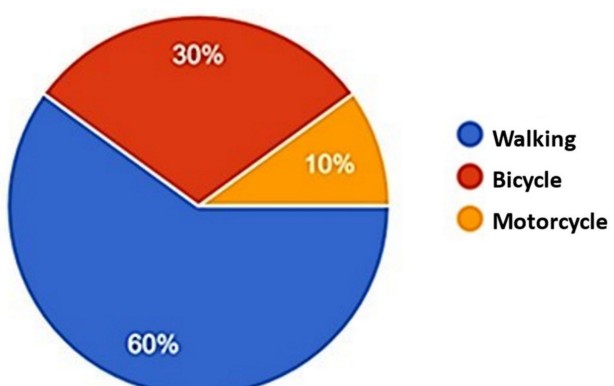

**Figure 5.** Means of transport used by teachers on their home–school journey.

*3.2. Teachers' Perspective on School and the Teaching Profession*

Regarding the perspective of teachers about the school and the teaching profession, several categories and subcategories emerged from the analysis of teachers' responses, which are summarized in Table 1.

**Table 1.** Analysis categories and subcategories.

| Categories | Subcategories | Category Description |
|---|---|---|
| The school and the teaching profession | I. The importance that the teacher attaches to his profession.<br>II. The constraints that the teacher feels in his profession.<br>III. The functions that the teacher is assigned at school.<br>IV. The importance of teacher training for their work.<br>V. What the school should look like, in the opinion of the teachers. | It reveals the vision that teachers have about the role and the aspects that they value in their profession.<br>It reveals the difficulties the teachers feel in their work.<br>It reveals the functions that the teachers are assigned at school.<br>It reveals if the training that teachers have received helps them in their work.<br>It reveals the vision that the teachers have of how their school should be. |

3.2.1. Importance of the Teaching Profession

As for the importance that the interviewed teachers attribute to their profession, some argued that teachers have a key role in society and in the formation of future adults, some of them with great social responsibility. One teacher said, "I also like being a teacher because being a teacher is very important. (. . .) if there are no schools you can't be the president, you can't be the doctor you can't work in a state because there's no school". Another teacher said, "I like to be a teacher because it is the key to the world".

Other teachers associated their work with the passion of teaching and the mission of transmitting knowledge to children: "I chose to be a teacher because the knowledge I have I need to transmit to other people"; "I like to teach, teach kids, teach what it is that I know". Some teachers also express pride in their profession, such as one of them who said, "I am very glad that there is something they have learned from me".

Some mentioned that they chose the profession because their father was also a teacher, i.e., "I also because I love being a teacher, because it is my father's profession, my father was also a teacher and he always tells me you are going to be a teacher", or because they wanted to be like their teachers: "Because I saw my teacher and also wanted to teach".

One of them spoke of the teacher as an agent of transformation of the students, referring mainly to the role of the teacher in the understanding that the student has of the knowledge and the world that surrounds them: "The teacher can transform the student, facilitate. . . to know a lot of things at school".

### 3.2.2. Constraints on Teachers' Work

The main constraints highlighted by the teachers were gathered around five themes: the Portuguese language, the lack of didactic materials, the lack of teachers, the isolation of the communities, and the lack of participation of the community in the school.

Some teachers referred to the difficulty they experience in teaching in Portuguese, despite it being the official language in Guinea-Bissau. Frequently, teachers must appeal to mother tongues so that students can understand the lesson and thus allow communication between students and teacher. The teachers report that even the Creole language is not spoken by the majority of children in their schools: "In our community many students do not know how to speak the Creole language as the Portuguese language, only the mother tongue"; "The boys if we speak Portuguese they don't learn, Creole more or less, just to speak mother tongues as soon as they learn".

Another difficulty that many teachers highlighted is the lack of teaching materials in schools; one of them said, "When the teacher has only one book, we must to write on the board, it takes a long time", and another teacher said, "that is why we always talk about school failure in Guinea-Bissau because of lack of didactic materials".

The teachers also referred to the isolation of the communities where they work, namely in the difficulty of transport and residence, because some of them are displaced from their homes. One of the teachers spoke about the difficulty they have in going to the training sessions due to lack of transport: "There are villages that did not have the cars (...) but I'll wait there for the car, but car only... sometimes I'm up to 10 or 11 am waiting for the car". Another teacher from the same school highlighted the daily effort they have to make to move from their residence to the school where they work: "I will reinforce my colleague, because we are colleagues from the same school. We lived far away, we left our village for school. Me...the distance between two villages is 7 km...this is also very, very difficult for me, because there is no residence for us to live in".

One of the teachers mentioned that the lack of teachers in their school is the main difficulty they feel in the exercise of their profession, explaining that they must work the morning shift and the afternoon shift. However, the teacher says it is an effort they must make because it was them that chose this profession. This teacher said, "lack of teachers... a teacher gives 1st, 2nd in the afternoon, there is nothing of subsidy, there is nothing, it is sacrifice because it is a task that we have chosen".

Some teachers associated the greatest difficulties in their profession with the lack of collaboration of the community and families, which was sometimes associated with the devaluation of the official school to the detriment of the Quranic schools: "the community does not want to know about the school"; "teachers do not receive money from the community"; "parents do not participate"; "In our village the Quranic school is taking the lead from the official school".

### 3.2.3. The Functions of the School

Regarding the vision that teachers have about the school, they referred to three main functions: the school as a key to the development of the community, the school as a place where knowledge is transmitted, and the school as a transmitter of values and behaviors.

Some teachers highlighted the role of the school in society, arguing that it is through the school that we can develop a society, saying that the "school is very important, because if there is no school in the community the community will not develop"; "The school is of great importance and also helps the children to participate in society".

Others highlighted the role of the school as a place of transmission of knowledge and where children move to learn, saying that "school is the home of children's learning"; "school is the place where everyone learns"; "I think school is of great importance because most of the children have left their home to come to school to learn from the teacher".

Other teachers referred to the school as a place of learning the values of society and the behavior that children should have in the community, referring to the following: "for me the school has great importance because a child who does not go to school has almost

a different behavior than the one who attended the school"; "School is a very important place for us. I think... We know how to learn through school. So, if we're teaching boys, tomorrow they'd be better men too".

### 3.2.4. The Importance of Teacher Education

As mentioned earlier, no teacher had initial pedagogical training before starting their work as a teacher. Thus, the majority reported that the continuous training they have received has helped a lot in their work in the classroom, both in dealing with children, in planning classes, and knowing better the content they teach.

Some teachers highlighted training as fundamental to the exercise of their work in the relationship that the teacher must have with the children and also in the formal aspects of the profession, such as writing the summary on the board or planning the lessons, saying that "the training we have allows us to work in school"; "The training we received... it makes it much easier for me in school because the training I receive here is this training that I develop in my school"; "I had difficulties at the beginning in writing the table of contents on the board, I didn't know how it was done and I had to do it at the beginning of each class. Through this training I learned that and more"; "In the beginning I had difficulties in dealing with the children, I did not have difficulties in terms of content, but with the relationship with children".

Other teachers said that it was through training that they learned to plan their lessons: "when I started teaching, I didn't know about lesson plans and other things, only with training I started to be able to work better"; "In this training we can relearn the way to plan, that the teacher has to plan his lesson".

Some teachers highlighted the role of training as a facilitator in learning the content they should address in the classroom, saying that "the information we received in the training gave us content that we are addressing... approach at school'; "Look! That training that we are doing here helps me a lot, because there are many subjects that are in the book...I forgot those subjects, but through that training I realize very much, very much".

One of the teachers said that their training helped them to be more creative in the use of local materials to teach their students, saying that "we always have problems of materials, but the teacher must be creative, in the aspect of arranging chopsticks or stones, all this our trainers taught us the way to use it".

### 3.2.5. What School Should Be Like

Regarding the conception that teachers have about how their school should be, most of the answers related to teachers asking for more infrastructure, more teaching materials, and greater community participation in the school, namely in the commitment to pay the subsidy they assume as teachers.

Some teachers began by mentioning that their school needs more basic infrastructure, such as access to water and latrines, saying that "In our school, in our tabanca we do not have water. At school there is no drinking water... It has no drinking water"; "Same situation also in my school. There is water but that water, If the water is put in a bottle until morning in the morning the bottle will turn red"; "The school doesn't have a latrine and that's why the children find it difficult"; "Latrine is also another issue that we have there, because the school almost situates in the middle of the village, the village almost goes around and school situates in the middle. Latrines are not okay. Nor the teacher when we need, we must leave school and go home".

Others also referred to the need to have a residence for teachers: "we don't have the residence, we need the residence"; "There is no teacher residence. The teacher should look for *djarga* (village chief) who can look for where the teacher should stay".

Some teachers highlighted the need to increase the infrastructure of the school, such as the number of classrooms and the creation of a room for pre-school education, noting that pre-school education assumes a great importance in preparing children for the 1st class.

One of the teachers said, "kindergarten is important because in our community there is no kindergarten. Directly to the first year of basic school. It takes a lot of patience for that child to be able to...first they must know how to write and read. But with a kindergarten it is very important"; "There we need a preschool class. So that students who have reached the first class, will have easier in terms of understanding. Imagine how I, this year, am working at the 1st level. There most students do not know the letter or know how to speak anything. We have a little bit of difficulty".

On the other hand, some teachers referred to the importance of community participation in school, arguing that "there should be the participation of all, especially the community", another reinforces with the question of the need to pay teachers "the community must participate and pay teachers".

Only one of the teachers referred to the need for teaching materials, saying that "we are experiencing a lack of materials (...) We also need from the beginning, these teaching materials in order to help in our work".

## 4. Conclusions

Community management schools in Guinea-Bissau are a reality present in much of the country, especially in the areas furthest from urban centers. These schools allow many children to have access to education, which would not otherwise be guaranteed. On the other hand, the teachers who work in these schools assume a decisive role in their functioning and in the relationship that the school has with the community where it operates.

None of the teachers participating in the study had initial pedagogical training before starting to exercise teaching functions. On the other hand, it was found that the continuous training of these teachers allowed them to perform their profession better, both in their tasks as a teacher and even in the knowledge of the content to be taught. It is therefore urgent to invest in education, in the definition of effective educational policies, in the creation of infrastructures, and in the training of professionals. Baldé [20] highlighted the importance of teacher training and the promotion of public policies focused on basic education, emphasizing that the lack of trained human resources is one of the main challenges for improving the quality of education in the country. The Government of Guinea-Bissau has been striving in recent years to strengthen the capacity for training teachers of basic education. Several international organizations also provide financial and technical support to strengthen and improve the capacity of the education system, notably through teacher training, construction and rehabilitation of schools, and provision of teaching materials.

Teachers have demonstrated that they attribute a fundamental role to the school in the development of the community and in the transmission of values to the children. As provided for the LBSE [2], basic education must "form, in freedom of conscience, civically responsible citizens and democratically involved in community life, providing students with experiences favorable to their civic and socio-affective maturity and the acquisition of autonomous attitudes". On the other hand, they also attribute to the school the function of a transmitter of knowledge.

The results evidenced are also in agreement with the opinion of Nóvoa [21], who stated that the vision of teachers about the school is built from their personal experiences as students and as education professionals. For these teachers, the school can be seen both as a space of opportunities, where we can learn, teach, and develop as a person, and as a place of frustration, where there are many difficulties, the lack of resources, and challenges in dealing with students.

In this study, no observation was made in the classroom, and the voices of the community and families were not heard. In any case, this case study allows us to contribute to the knowledge about the vision of teachers about the school and its profession, in particular for the teachers at rural community management schools in the eastern part of Guinea-Bissau.

In summary, it is important to improve teaching conditions, adequately remunerate teachers, and stabilize the political system to ensure the success of the education system [6].



Basic education in Guinea-Bissau needs significant investment to improve access to and the quality of education. It is necessary for the civil society and the state to work together to overcome existing obstacles and promote significant changes in the Guinean education system. It is important that, in the future, continuing research can include the voices of the different actors in the educational community so that, in a more in-depth way, the vision that teachers have of the role of the school and its profession in society is understood since this is a complex subject that involves several factors and aspects. Finally, we consider it essential that studies be carried out to better understand this theme and so that educational policies can be implemented that value and encourage the training of committed and motivated teachers for their activity.

**Author Contributions:** Methodology, F.D. and M.J.R.; Investigation, F.D. and M.J.R.; Data curation, F.D.; Writing—Original draft, F.D. and M.J.R.; Writing—Review & editing, M.J.R. All authors have read and agreed to the published version of the manuscript.

**Funding:** This work has been supported by FCT—Fundação para a Ciência e Tecnologia within the Project Scope: UIDB/05777/2020.

**Informed Consent Statement:** Informed consent was obtained from all subjects involved in the study.

**Data Availability Statement:** Not applicable.

**Conflicts of Interest:** The authors declare no conflict of interest.

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
