# Peer review of "The Voice of Teachers about School and the Teaching Profession in Guinea-Bissau"

_education, doi:10.3390/educsci13080802_

Round 1

Reviewer 1 Report

This study examines teachers’ vision of the school and the teaching profession. It is interesting and important to know how teachers perceive their school. I have the following suggestions for improving the paper:

1.     Include some literature about how teachers perceive education or their school. The current paper gives us a detailed background of the educational system in Guinea-Bissau, which is very good. But, I think the authors should also do a systematic review about previous literature in this field.

2.     The authors can also add a table to give us some basic information about the interviewees, such as their gender, age, years of working experience, etc.

3.     For the findings, I suggest the authors to improve the data analysis. For example, in the section of “Importance of the teaching profession”, the authors can divide this section into several parts regarding different ideas about the importance of teaching profession. The data should be presented in a more systematic way with stronger logic. Besides, different sections of the finding are not closely connected.

4.     For the finding about the functions of the school, did any teacher mention the role of school in improving students’ social mobility?

To conclude, I think this is an interesting and valuable study. But improvements are needed.

It is okay. 

Author Response

  1. Unfortunately, there is not much recent reference literature about the educational system in Guinea-Bissau. The one that is most relevant and that we managed to find, we put it.
  2. The characterization of teachers is described in the text and in graphics in a simple and clear way. The creation of a table with this data will create repetition in the presentation of the results.

  3. These sections are presented according to the results obtained during the interviews, justified in the table that describes the analysis subcategories.
  4. This was not mentioned and, for that reason, is not present in the results.

Reviewer 2 Report

This article uses a case study to study the knowledge about the vision of teachers about the school and its profession, in particular the teachers of rural community management schools, in the eastern part of Guinea-Bissau. Although this case can help us understand the role that teacher training plays in the quality of their work and the efforts necessary for the school to respond to the needs of society in Guinea-Bissau. However, the research method uses case studies, which is highly subjective, and the conclusions obtained are not general and have no reference significance for the education development in other regions. In addition, although this research has some practical value, it lacks theoretical basis and poor academic rationality. 

In addition, the study has the following problems:First of all, the introduction does not point out why you study the issue of educational development in Guinea-Bissau, what is the significance? In addition, the importance of studying teachers' views on schools is not pointed out, that is, the research context is incomplete, and the research significance of the topic is not pointed out.

In Materials and Methods, what are the reasons for choosing this sample, and are the conclusions from this sample survey representative? Can the recommendations made in this region be extended to other regions ? "Content analysis involved a subjective process, subject to the  interpretation of researchers", so is it heavily influenced by the subjective nature of the content analysis, and how representative and general are the conclusions you get from that?

Third, the survey of teachers' views on schools only accounts for a part of the survey content. It is suggested that the author replace the topic with the vision of teachers about the school and its profession, rather than only the views on schools

Finally, based on the research content, you briefly described your research results without analyzing them

Minor editing of English language required

Author Response

This group of teachers was chosen because it is a group of teachers that is part of a network of community management schools supported by an international NGO. At the same time, these teachers have biweekly meetings for training and planning classes, which facilitated the formation of focus groups for interviews. The fact that the group of teachers belong to a network of schools and meet every fortnight in one place facilitated access to teachers and data collection. In a case study it is not intended to generalize the results obtained, but on the contrary to understand in depth a certain phenomenon that allows to contribute to the knowledge that one has on this matter.

We changed the title to include two topics: the vision of the school and the vision of the teaching profession.

With this case study we intend to describe the vision that teachers have about the school and their profession. We understand that the description of the data obtained, by itself, already lacks an analysis, where we distribute them by several subcategories and where we present them in a systematic and organized way. In a case study it is intended to deepen the knowledge about a specific case, it is not intended to explain or generalize the results found.

Reviewer 3 Report

Based on semi-structured interviews with 30 elementary schools in Guinea-Bissau, the paper provides insights into how teachers operate within and view the school system. The results show that, while most teachers share the belief in the value of their work and education of pupils, their work is hindered by a variety of "constraints," such as infrastructural challenges of the school facilities, low pay, lack of teaching materials, etc. Based on these findings, the authors make recommendations for systemic improvements that could be taken on by the school system to provide better conditions for teaching and learning. 

I find the focus on teachers' perceptions adequate, while a more comprehensive study involving students, parents, and community members would be much beneficial to the literature.

However, I do feel that the paper lacks some literature review and historic/geographic context. I hoped that there would be some context about whether similar conditions are experienced in similar geographies. Or literature on the challenges of schooling in resource-poor areas. The paper also left us wondering about the background of education in Guinea-Bissau. There is some information about the structuring of education, but there is no historic context to how and why the system was shaped as such. Given that there is quite a bit of emphasis on the school facility as a locus of social improvement and transformation, it would also be helpful see at least one or two examples of what these schools actually look like.

I would also suggest that authors consider providing examples of improvement that may have been taken on by similar education systems or even within G-B school system itself. In the way it is presented, there are obvious recommendations to how to improve the schools. It is obvious that better sanitation, accessible teaching materials, and better teacher training would help matters. To be sure, more resources will lead to better schools. But can the authors tease out a path or a set of priorities given that such resources may not be readily available or available all at once? 

I would reconsider the sentences that suggest teachers are only men. Currently they are overwhelmingly men but consider using they, instead of he. 

In one instance, there is a typo: mut should read must.

Author Response

We have added a brief description of the schools in the methodology chapter.

We have made the corrections suggested in the text.

Reviewer 4 Report

Suggestions for the theoretical part:

- It is suggested in the title after "school" to put "and teacher profession", because the paper and the research deal with both themes.

- Since the research includes case study, it is necessary to supplement the description of the context of the case, i.e. legal regulations and/or other formal documents and information that describe how the teacher profession and/or the school system in Guinea-Bissau in general are regulated.

Suggestions for the research design:

- In chapter 2 a more clear presentation of the research design is needed: there should be research questions added. That is also possible by rephrasing the penultimate paragraph in the Introduction. Accordingly it is suggested to rename the chapter 2 by using a more general phrase, i. e. "The Research Methodology".

- The method of sampling should be clarified, more precisely on what criteria the 30 teacher were chosen.

- On page 3, after "characteristics" of the teachers it is suggested in parentheses to list these characteristics.

- Short explanation why for the research the case study was chosen and why the focus group in this context. It should be also mentioned how many focus groups were organized and with how much participants. 

- If possible, in the Results or Conclusions should be shown more connection with theory and/or earlier similar research about this topic (some already mentioned in the Introduction). In this regard it is suggested to better clarify and argue the teacher statements in each category. The interpretation also needs better explanation, in which way the results could contribute to the improvement of education policy. Is it meant only for Guinea-Bissau or broader?

Author Response

We've changed the title to be more complete, as suggested.

We have made the suggested changes in the text.

This group of teachers was chosen because it is a group of teachers that is part of a network of community management schools supported by an international NGO. At the same time, these teachers have biweekly meetings for training and planning classes, which facilitated the formation of focus groups for interviews. The fact that the group of teachers belong to a network of schools and meet every fortnight in one place facilitated access to teachers and data collection. In a case study it is not intended to generalize the results obtained, but on the contrary to understand in depth a certain phenomenon that allows to contribute to the knowledge that one has on this matter.

Round 2

Reviewer 2 Report

Thank you for the revisions. 

Minor editing of English language required

Author Response

Thank you for your suggestions.

Reviewer 3 Report

Thank you for the revisions. For the future work, it would be great to have photos to give a better sense of the context.

There are still several typos to pick up in instances such as "they works."

The revised title is more specific but you may consider the following option to simplify it:

"The voice of teachers about school and teaching"

OR

"The voice of teachers about school and teaching profession"

Please see my notes above, some of the revised sentences need to be edited.

Author Response

Thank you for your suggestions. We reviewed the english issues. 

Also we update the title to simplify it. 

Thank you